# Synergistic Effects of *Clonostachys rosea* Isolates and Succinate Dehydrogenase Inhibitors Fungicides against Gray Mold on Tomato

**DOI:** 10.3390/microorganisms11010020

**Published:** 2022-12-21

**Authors:** Jiehui Song, Tengyu Lei, Xiaojuan Hao, Huizhu Yuan, Wei Sun, Shuning Chen

**Affiliations:** 1Key Laboratory of Pesticides Evaluation, Ministry of Agriculture, Institute of Plant Protection, Chinese Academy of Agricultural Sciences, Beijing 100193, China; 2Jiangsu Key Laboratory of Crop Genetics and Physiology & Co-Innovation Center for Modern Production Technology of Grain Crops, Agricultural College, Yangzhou University, Yangzhou 225009, China; 3College of Agriculture, Shanxi Agricultural University, Jinzhong 030801, China

**Keywords:** tomato gray mold, *Clonostachys rosea*, succinate dehydrogenase inhibitors (SDHI), synergistic effect

## Abstract

Gray mold caused by *Botrytis cinerea* is a devastating disease in tomatoes. Site-specific fungicide application is still key to disease management; however, chemical control has many drawbacks. Here, the combined application of a biological agent, *Clonostachys rosea*, with newly developed succinate dehydrogenase inhibitors (SDHI) fungicides showed stronger synergistic effects than the application of SDHI fungicides alone on tomato gray mold control. *C. rosea* 67-1 has been reported as an efficient biological control agent (BCA) for *B. cinerea*. Little information is currently available about the combination of *C. rosea* and fungicides in the control of gray mold. By testing the sensitivity to fungicides with different action mechanisms, *C. rosea* isolates showed high tolerance to SDHI fungicides (1000 μg mL^−1^) on PDA, and the conidial germination rate was almost not affected under 120 μg mL^−1^ of fluxapyroxad and fluopyram. In greenhouse experiments, the control effect of the combination of *C. rosea* and fluxapyroxad or fluopyram against tomato gray mold was significantly increased than the application of BCA or SDHI fungicides alone, and the combination allows a two-fold reduction of both the fungicide and BCA dose. Further, the biomass of *B. cinerea* and *C. rosea* on tomato plants was determined by qPCR. For *B. cinerea*, the trend of detection level for different treatments was consistent with that of the pot experiments, and the lowest biomass of *B. cinerea* was found when treated with *C. rosea* combined with fluxapyroxad and fluopyram, respectively. For *C. rosea*, qPCR assay confirmed its colonization on tomato plants when mixed with fluopyram and fluxapyroxad. These results indicated that combining *C. rosea* 67-1 with the SDHI fungicides could synergistically increase control efficacy against tomato gray mold.

## 1. Introduction

Gray mold caused by *Botrytis cinerea* can be a devastating disease in tomatoes worldwide. It is also common with numerous other fruit, vegetables, and ornamental crops [1], which makes it difficult to control. Although cultural methods such as appropriate plant spacing, rational fertilization, and breeding disease-resistant varieties can reduce disease incidence, site-specific fungicide application is still crucial to disease management [2]. However, the polycyclic nature of the disease, abundant sporulation, high genetic variability, and short generation time of the pathogen contribute to a high risk for the development of resistance to site-specific fungicides used for control [3]. Several of the most serious issues of fungicide resistance have been reported in *B. cinerea*, including resistance to methyl benzimidazole carbamates, dicarboximides, succinate dehydrogenase inhibitors (SDHI), anilinopyrimidines, quinone outside inhibitors, phenylpyrroles, and sterol biosynthesis inhibitor class III fungicide hydroxyanilide, etc [4,5,6]. Besides the resistance risk, chemical control has certain drawbacks, such as phytotoxicity to tomato plants, toxicity to non-target organisms, and stringent requirements for correctly timing the fungicide application, which all hinder its usage and ability to effectively control disease [7,8].

The application of biological control agents (BCAs) to manage tomato gray mold is a promising alternative to synthetic fungicides [9,10,11]. Among them, *C. rosea* has been shown to be effective in controlling gray mold in several crops, both in field and greenhouse cultivations [12,13,14,15]. It protects plants against *B. cinerea* by inhibiting spore production and suppressing gray mold development [16]. The defense mechanisms of tomato plants against gray mold, including changes in the signaling molecule and defense enzyme activity, could also be induced when treated with *C. rosea* [16,17,18,19]. In recent years, *C. rosea* has been commercially available as a biofungicide, which can effectively control many plant diseases, including gray mold, and has been applied to millions of hectares in China. Although BCA may decrease the frequency and total amount of fungicide spraying, reducing residues and resistance risk, their effectiveness is usually inferior to that of chemical fungicides because of the complexity of the field environment [11,20]. Given the limitations of biological and chemical control strategies, combining *C. rosea* with newly developed fungicides may develop a practical method to control *B. cinerea* in tomato fields. Many studies reported the combination of BCAs and fungicides in controlling plant disease. However, few studies showed the combination of *C. rosea* and fungicides to control plant disease.

The SDHIs are the group that rapidly incorporates new broad-spectrum compounds in the market [21]. They have the specific function of preventing mitochondrial respiration by inhibiting the activity of mitochondrial respiration complex II, which consists of a flavoprotein (SdhA), ferritin (SdhB), and two membrane anchoring proteins (SdhC and SdhD) [22]. Regardless of the SDHI high fungicides efficacy, these fungicides are classified as the medium-to-high risk of resistance [23,24]. Resistance to carboxin, boscalid, penthiopyrad, and fluopyram was reported shortly after their registration [21,23,25,26,27,28,29]. Thus, resistance management practices, such as rotation with different FRAC code fungicides and reducing the rates of fungicide application, must be implemented for the sustained efficacy of SDHI fungicides against the gray mold of tomatoes. 

*C. rosea* has been reported as an efficient biological control agent for *B. cinerea*. Its combination with fungicides may prolong the fungicides’ life and provide a viable strategy for disease control. Thus, this study aimed to (1) evaluate the compatibility of the SDHI fungicides and the antagonistic isolate *C. rosea* and (2) determine the synergistic effect of combined application of SDHI fungicides fluxapyroxad as well as fluopyram with *C. rosea* 67-1 for control of tomato gray mold in the greenhouse. 

## 2. Materials and Methods

### 2.1. Fungal Isolates and Pesticide

The *C. rosea* isolates (JLB-7-1, 67-1, SYP-4-2, SHW-1-1, YJS-3-2, GS6-1, NHH-48-2, BD-2-1) were provided by the Manhong Sun’ lab. Among them, isolate 67-1 has been reported in previous studies [30,31]. *B. cinerea* isolates YN80 and YN81 were collected from a tomato from Yunnan province. Isolates were recovered from stock cultures stored with silica blue gel beans at −20 °C on dried filter paper discs (Fisher Scientific, Pittsburgh, PA, USA). All isolates used for inoculations were maintained on PDA (Potato Dextrose Agar: potato 200 g L^−1^, glucose 15 g L^−1^, agar 15 g L^−1^, add deionized water to 1 L. Fresh potato was boiled in deionized water for 20 min, then filtered and the potato juice were taken to make medium) medium at 25 °C in darkness unless otherwise specified. The isolates from storage were grown for five days on PDA before being used for experiments.

Technical-grade carbendazim (98% a.i;Jiangsu Longdeng Chemical Co., Ltd., Suzhou, China), 96% tebuconazole (Guangxi Nanning Guangphthalein Agricultural Chemical Co., Ltd., Nanning, China), 98% pyraclostrobin (Shaanxi Diedu Medichem Co. Ltd., Xian, China), 97% Boscalid (Beijing Bailingwei Technology Co., Ltd., Beijing, China), 98% fluxapyroxad (BASF Corp., Research Triangle Park, NC, USA), 98% fluopimomide (Shandong Zhongnong United Biotechnology Co., Ltd., Jinan, China), and 98% fluopyram (ACMEC, Shanghai, China) were used in this study. Stock solutions were made by dissolving each fungicide in DMSO at the concentration of 10^5^ µg a.i. mL^−1^. The stock solutions were stored at 4 °C in darkness. Salicylhydroxamic acid (SHAM, 99% a.i.; Syngenta Biotechnology Co. Ltd., Shanghai, China) was added to pyraclostrobin-amended PDA at 100 µg mL^−1^ to suppress the alternative oxidase pathway [32]. Corresponding control dishes contained SHAM.

### 2.2. Fungicides Sensitivity Assessments of C. rosea and B. cinerea In Vitro

Sensitivity to carbendazim, tebuconazole, boscalid, and pyraclostrobin was assessed on fungicide-amended PDA at 0, 0.1, 0.3, 1, 3, 10, and 30 µg a.i. mL^−1^. Furthermore, sensitivity to boscalid, fluxapyroxad, fluopimomide, and fluopyram was assessed on fungicide-amended PDA at 0, 0.1, 0.3, 1, 3, 10, 30, 100, 300, 1000, and 3000 µg a.i. mL^−1^. To inoculate test plates, mycelial plugs were removed with a 5-mm cork borer from the margins of 5-day-old colonies and placed upside down on the centers of 9-cm plastic Petri dishes containing fungicide-amended or unamended media. Each isolate was tested in triplicate, and plates were incubated until the diameter reached 60 mm (around five days for *B. cinerea* and nine days for *C. *rosea**). Fungicide sensitivity, as measured by the 50% effective concentration (EC_50_) value, was calculated as described by Wong and Wilcox (2002) [33]. Briefly, the percent relative growth (RG) was calculated as (radial growth at fungicide concentration/radial growth on the non-amended control plate) × 100. The EC_50_ value was estimated by linear regression of the probit-transformed relative inhibition (RI) value (RI = 1 – RG) on log10 transformed-fungicide concentration. The EC_50_ value for each isolate was calculated as the mean of the three replicates.

### 2.3. Effect of SDHI Fungicides to C. rosea Conidia Germination

To determine the inhibition effect of SDHI fungicides boscalid, fluxapyroxad, fluopimomide, and fluopyram on *C. rosea* and *B. cinerea*, a spore germination rate test was conducted as described. To stimulate sporulation, *C. rosea* isolate 67-1 was inoculated in Czapek Dox Liquid Medium (Sigma-Aldrich, St.Louis, MO, USA) [34]. *B. cinerea* isolate YN80 was inoculated in a PDA medium. Conidia were harvested by flooding 1–2-week-old *C. rosea* and *B. cinerea* cultures with a sterile scraper and suspending them in sterile distilled water. The conidial concentration of *C. rosea* and B. cinerea was then quantified microscopically using a hemocytometer and diluted to a concentration of 1.0 × 10^6^ conidia mL^−1^. An aliquot of 200 µL of conidia suspension was plated on the YBA medium (10 g L^−1^ bacto-peptone (Sinopharm, Beijing, China), and 20 g L^−1^ sodium acetate (Sinopharm, Beijing, China), 10 g L^−1^ yeast extract (Sinopharm, Beijing, China), and 15 g L^−1^ agar (Sinopharm, Beijing, China)), then mixed with fungicide using a sterile glass spreader at the final concentrations of 0, 7.5, 15, 30, 60, and 120 µg mL^−1^. After 18–24 h incubation at 25 °C in the dark, the number germinated per 100 conidia was counted, and the germination rate of conidia was calculated. The experiment was performed twice.

### 2.4. Greenhouse Experiments

Tomato (*Solanum lycopersicum* Dunal L.) seedlings (Jinpengwuxian, Xi’an Jinpeng Seedling Co., Ltd., Xi’an, China) were planted in 1 kg of autoclaved potting medium (field soil/peat/sand, 1:1:1 wt/vol/wt; one seedling per pot) and maintained under a 16-h photoperiod at 90% relative humidity and 25 °C room temperature. Forty-day-old tomato seedlings were used for the inoculation test. Nine treatments were applied to the seedlings to measure the synergistic effects of *C. rosea* and SDHI fungicides: 1. YN80 treatment, inoculated with mycelial plugs of *B. cinerea* isolate YN80 and sprayed with distilled water; 2. 67-1 treatment, sprayed with 6 mL of 10^7^ conidia mL^−1^ conidia suspension of *C. rosea* isolate 67-1; 3. fluxapyroxad treatment, sprayed with 6 mL of 30 μg mL^−1^ a.i. fluxapyroxad; 4. fluopyram treatment, sprayed with 6 mL of 30 μg mL^−1^ a.i. fluopyram; 5. 67-1 combined with fluxapyroxad treatment, sprayed with the mixture of conidia suspension and fluxapyroxad (5 × 10^6^ conidia mL^−1^ conidia suspension: 15 μg mL^−1^ a.i fluxapyroxad, 1:1); 6. 67-1 combined with fluopyram treatment, sprayed with the mixture of conidia suspension and fluopyram (5 × 10^6^ conidia mL^−1^ conidial suspension:15 μg mL^−1^ a.i fluopyram, 1:1); 7. 67-1 rotate with fluxapyroxad treatment, sprayed the 5 × 10^6^ conidia mL^−1^ conidia suspension first, and fluxapyroxad (15 μg mL^−1^ a.i) 24 h later; 8. 67-1 rotate with fluopyram treatment, sprayed the 5 × 10^6^ conidia mL^−1^ conidia suspension first, and fluopyram (15 μg mL^−1^ a.i) 24 h later; 9. blank control, only sprayed with 8 mL of distilled water. Moreover, 0.1% Tween 80 was included in all spray treatments as a surfactant. 

After 24 h, all of the above tomato seedlings treatments were inoculated with 5-mm-agar plugs of *B. cinerea* isolate YN80 on the leaves referred to Myresiotis et al. [32], except the blank control treatment. Each plant was inoculated with ten agar plugs, one plug for each leaf. Six pots were prepared for each treatment. After inoculation, tomato plants were immediately returned to the chamber to maintain a high relative humidity and an appropriate temperature. Seven days after inoculation, lesion diameters were measured at two perpendicular directions using a caliper, and the control efficacy of each treatment was calculated. The experiments were performed three times. 

### 2.5. qPCR for Specific Quantification of C. rosea and B. cinerea

To measure the concentration of the DNA, standard plasmids were constructed. The DNA sequence for *B. cinerea* was amplified using the primers P1 (5′-GCTGTAATTTCAATGTGCAGAATCC-3′) and P2 (5′-GGAGCAACAATTAATCGCATTTC-3′) targeting the Bcos5 gene as reported by Duan et al. [35]. As for *C. rosea*, primers targeting β-tubulin-encoding genes were retrieved from Genbank (Accession number AF435066). Primers CLO-QF/CLO-QR (CAACAACAACGAGTGGGGAG/ATAAAAGACGGAGCGAAGAC) were designed and used in this study. PCR reactions were performed as follows: 95 °C for 5 min, and then 35 cycles of denaturation at 95 °C for 30 s, annealing at 60 °C for 30 s, extension at 72 °C for 30 s, with a final extension at 72 °C for 10 min. Then, purified PCR products were inserted into the cloning vector pClone007 Vector Kit (Tsingke Biotechnology, Beijing, China), and transformed into an *E. coli* DH5α competent cell. The transformed competent cells were coated in the LB medium (Luria-Bertani: tryptone (Sinopharm, Beijing) 10 g L^-1^, yeast extract (Sinopharm, Beijing) 5 g L^−1^, NaCl (Sinopharm, Beijing) 10 g L^−1^, agar (Sinopharm, Beijing) 15 g L^−1^) containing 200 μg mL^−1^ of ampicillin, and incubated at 37 °C to obtain the target cell after 12–16 h. The plasmid DNA was extracted from the target cell using a plasmid mini kit (Tsingke Biotechnology, Beijing, China). The plasmid DNA was used for preparing 10-fold dilution series of eight concentration points starting with about 10 ng/μL, as a “fungal DNA series”. The initial stock solution contained around 3 × 10^8^ target copies/μL, which was calculated by converting the stock concentration and the mass of the fragment into copy numbers. The concentration of plasmid DNA was quantified by spectrophotometry. The standard curve was prepared in fungal DNA series and amplified to obtain standard curves. Each standard curve was measured in three technical replicates. Standard curves were generated by plotting the logarithmic values of target copies versus the corresponding cycle threshold (Ct) values and fitted into a linear regression model. It was always checked that the R^2^ of standard curves ranged from 0.99 to 1. Only Ct values inferior to 40 for *B. cinerea* and 35 for *C. rosea* were considered to avoid false positives, and each standard was measured in three technical replicates.

Following the method in Section 2.4, fifteen leaves (five for each plant) were collected from treatment “*B. cinerea* treatment”, “*C. rosea* treatment”, “fluxapyroxad”, “fluopyram”, “67-1 combined with fluxapyroxad treatment”, “67-1 combined with fluopyram treatment” and then ground into a fine powder under liquid nitrogen. For each sample, 150 ± 2 mg was used for DNA extraction to detect fungal content by qPCR. The genomic DNA was subsequently extracted using the Plant Genomic DNA Kit (TIANMO BIOTECH, Beijing, China) according to the manufacturer’s instructions.

As for the qPCR detection of *B. cinerea* and *C. rosea,* primers P1/ P2 and CLO-QF/CLO-QR for the construction of standard plasmid were used. All qPCR reactions were performed on QuantStudio™ 6 Flex Real-Time PCR System (Applied Biosystems, Waltham, MA, USA) in transparent Multiwell 96-well plates and sealed with adhesive foil. Twenty micro-liter reaction volume contained 10 μL TSINGKE TSE201 2×TSINGKE^®^ Master qPCR Mix (SYBR Green I) (Tsingke Biotechnology Co., Ltd., China), 0.8 μL of each primer, 0.4 μL 50×ROX Reference Dye II (Tsingke Biotechnology Co., Ltd., China), 7 μL of DNAse-free water, and 1 μL of DNA sample (unless otherwise stated). The detection wavelength was 520 nm ±10 nm. The following thermal program was applied: an initial denaturation step of 94 °C for 5 min, followed by 40 amplification cycles of 15 s denaturing step (94 °C) and 60 s annealing-extension step (60 °C). All of the experiments were repeated independently twice. Three replications per sample were included in all of the experiments. 

### 2.6. Statistical Analysis

Control efficacy = [(lesion diameter of the control − lesion diameter of the treatment)/lesion diameter of the control] × 100%. Results were represented as the mean values ± standard deviation. One-way analysis of variance (ANOVA) with a least significant difference (LSD) test in SPSS software (version 21.0; IBM SPSS Inc. Chicago, IL, USA) was used to evaluate the significant differences between treatments. 

## 3. Results

### 3.1. In Vitro Mycelial Growth Inhibition of C. rosea and B. cinerea by Differernt Fungicides

To test the compatibility of *C. rosea* and fungicides, carbendazim, tebuconazole, pyraclostrobin, and boscalid were selected as representative fungicides for Methyl Benzimidazole Carbamates (MBCs), sterol demethylation inhibitors (DMIs), quinone outside inhibitors (QoIs), and SDHIs fungicides, respectively. The sensitivity of *C. rosea* isolates to those fungicides was tested (Figure 1). Overall, *C. rosea* isolates displayed the strongest tolerance to SDHI fungicide boscalid. Boscalid at 10 μg mL^−1^ or 30 μg mL^−1^ showed no suppressive activity against mycelium growth of *C. rosea* on PDA medium. In contrast, *C. rosea* isolates were quite sensitive to cabendazim and pyraclostrobin, with EC_50_ values of 0.34 μg mL^−1^-1.66 μg mL^−1^ and 0.52 μg mL^−1^-11.17 mL^−1^, respectively. Tebuconazole also had an inhibitory effect on *C. rosea* mycelia for most of the isolates tested (except for isolate NHH−48-2), with EC_50_ values of 0.02 μg mL^−1^-21.11 μg mL^−1^. *C. rosea* isolate NHH-48-2 was tolerant to tebuconazole, with EC_50_ values of 102.86 μg mL^−1^ (Table 1).

To further explore the compatibility of SDHI fungicides with *C. rosea* isolates, more fungicides from the same categories were tested for their effects on *C. rosea* isolates. A more comprehensive range of concentration was tested for SDHI fungicides boscalid, fluxapyroxad, fluopimomide, and fluopyram from 0.1 μg mL^−1^ to 3000 μg mL^−1^. All tested *C. rosea* isolates displayed strong tolerance to all SDHI fungicides tested. When treated with 100 μg mL^−1^ of SDHIs, the growth of mycelium was only suppressed by 9.11% to 28.20% (Figure 2). Even when treated with 3000 μg mL^−1^ of SDHIs, the mycelium could grow by 53.73% to 77.96% compared to the unamended control. In contrast, the *B. cinerea* isolates YN80 and YN81 were sensitive to all the SDHI fungicides tested, with EC_50_ less than 15.46 μg mL^−1^ (Table 1). Based on the EC_50_ value, fluxapyroxad and fluopyram were most effective against the *B. cinerea* isolates used in this study. Thus, those two fungicides were selected for the following experiments.

### 3.2. Inhibition Effect of Fungicides on the Germination Rate of C. rosea Conidium

The germination inhibition assays of SDHI fungicides were also conducted in our study. The SDHI fungicides had strong inhibitory activity on the spore germination of *B. cinerea*. The germination rate of YN80 was less than 10% when treated with 15 μg mL^-1^ of fluxapyroxad and fluopyram (Table 2). In contrast, the inhibitory activity of fluxapyroxad and fluopyram against *C. rosea* was very weak. A strong residual growth (with a germination rate above 95%) was observed for *C. rosea* isolate 67-1 when treated with 120 μg mL^−1^ of fluxapyroxad and fluopyram. Thus, good compatibility was observed for SDHI fungicides and *C. rosea* in vitro.

### 3.3. Synergistic Effects of C. rosea Isolate 67-1 and SDHI Fungicides against Tomato Gray Mold in the Greenhouse

The data regarding the combined effects of *C. rosea* isolate 67-1 and SDHI fungicides against tomato gray mold in the greenhouse are presented in Figure 3 and Table 3. The average disease diameter in the control group was 2.67 cm in the greenhouse, indicating that *B. cinerea* was successfully inoculated and well developed (Figure 3). Overall, the combined application of *C. rosea* and SDHI fungicides, either in a mixture or in a rotation, significantly reduced the disease incidence and severity of tomato gray mold. The highest control efficacy of 77.07% was obtained with pretreatment of isolate 67-1 at 5 × 10^6^ conidia mL^−1^ and then fluopyram at 15 μg mL^−1^. The control efficacy of the combined application of isolate 67-1 with fluxapyroxad and fluopyram reached 70.91% and 71.94%, respectively. Sole treatment of fluxapyroxad and fluopyram at 30 μg mL^−1^ produced a significantly lower control efficacy of 52.28% and 58.31%, respectively, while *C. rosea* treatment 10^7^ conidia mL^−1^ yielded a control efficacy of 46.42% (Table 3).

### 3.4. qPCR for Specific Quantification of C. rosea and B. cinerea

The evaluation of the Ct values from the standard curve amplification for both *B. cinerea* and *C. rosea* revealed a linear dynamic range from 10^2^ to 10^6^ target copies, corresponding to a Ct range of 39~14 for *B. cinerea* and 32~14 for *C. rosea* (Figure 4). The lower limit of detection of *C. rosea* was determined around one target copy per reaction as 35 cycles were set to be the cutoff value for the method. Similarly, 40 cycles were set to be the cutoff value for *B. cinerea*. Linear regressions between the log-transformed number of target copies and the corresponding Ct values revealed R^2^ values > 0.99 for both *B. cinerea* and *C. rosea* reactions. No PCR inhibition was observed when different amounts of plant DNA isolated from tomato plants were added to the qPCR, increasing concentrations from 1, 10, 25, 50, to 100 ng (data not shown).

Thus, the qPCR method was applied to determine the survival of *B. cinerea* and *C. rosea* on tomato plants. In sample sets, *B. cinerea* and *C. rosea* were always detected when applied and not in the negative control samples. For *B. cinerea*, the trend of detection level for different treatments was inconsistent with those in the pot experiments. Take repeat 1, for example: two of the lowest levels of detection, with 1.66 × 10^4^ copies and 1.08 × 10^4^ copies, reflecting the lowest survival of *B. cinerea*, were found when treated with *C. rosea* combined with fluxapyroxad and fluopyram, respectively. When treated with *C. rosea,* the detection levels (with copies of 6.37 × 10^4^) were higher than those that were treated with fluopyram or fluxapyroxad (with copies of 3.72 × 10^4^ and 3.69 × 10^4^, respectively) but lower than those that were treated with distilled water (with copies of 8.55 × 10^4^). For *C. rosea*, the qPCR results showed that *C. rosea* could still be detected on tomato plants when mixed with fluopyram and fluxapyroxad (Figure 5).

## 4. Discussion

A combination of synthetic fungicides with BCA or a combination of different BCAs has been reported to reduce chemical application rates. Several combinations of BCA with fungicides have shown greater efficacy than the individual treatments. For example, combining *B. amyloliquefaciens* SDTB009 with difenoconazole is an effective strategy for tomato *Fusarium wilt* management [8]. Synergistic effects have been observed in the combined application of *Bacillus subtilis* H158 and strobilurins for rice sheath blight control [36]. The combination of *Trichoderma* and hymexazol enhanced antagonistic effects towards *F. oxysporum* [37]. Besides, the combination of *Metarhizium robertsii* and *Trichoderma asperellum* reduced the malathion doses in controlling ambrosia beetles [38]. However, few studies showed the combination of *C. rosea* and fungicides or other BCAs in the control of plant disease. In this study, the compatibility of SDHIs fungicides was evaluated and the synergistic effect of the combined use of *C. rosea* and SDHI fungicides against tomato gray mold was investigated. 

The action targets of fungicides against pathogenic fungi include cell membrane integrity, cell mitosis, nucleic acid metabolism, respiration, signal transduction, and protein synthesis [24]. However, some active ingredients of fungicides also act on non-target or beneficial microorganisms such as BCAs, which reduce the growth and population size of BCAs and limit the biocontrol effect [39]. Therefore, knowledge of the compatibility of BCAs and fungicides is essential to allow combined applications. Generally, fungal BCAs resistant to specific fungicides or bacterial BCAs have good compatibility. Compared with the biocontrol fungus, biocontrol bacteria, such as *B. amyloliquefaciens* and *B. subtillis* have been reported to tolerate many fungicides and exhibit synergistic effects when applied in combination [40,41,42,43]. The combination of hymexazol-resistant *Trichoderma* isolate with hymexazol also showed good compatibility and enhanced antagonistic potential [37]. Potential additive or synergistic effects of *C. rosea* and fungicides depend first on the biological compatibility between the biocontrol agent and the synthetic chemical. In this study, we screened several different categories of fungicides to identify their compatibility with *C. rosea*. Four FRAC code fungicides that are frequently used for the control of gray mold have been selected. *C. rosea* isolates were quite sensitive to carbendazim, pyraclostrobin, and tebuconazole in vitro. Fortunately, we found that *C. rosea* could tolerate SDHI fungicides, including boscalid, fluxapyroxad, fluopimomide, and fluopyram. Even when treated with 3000 μg mL^−1^ of SDHIs, the mycelium could grow quite well. The natural resistance of fungus to SDHI fungicides are not uncommon. The insensitivity of plant pathogens *Colletotrichum* species to boscalid, fluxapyroxad, and fluopyram have been confirmed on media and on plants [44]. Penflupen, a novel SDHI fungicide, exhibited good bioactivity against *F. fujikuroi*, but weak activity against other *Fusarium* spp. [45]. So far, the inherent resistance mechanisms in the above plant pathogens have remained unknown. As for *C. rosea*, the natural resistance to SDHIs allows them to be mixed with fungicides. 

*C. rosea* 67-1 isolate has been reported to be a highly efficient biocontrol fungus targeting many plant pathogenic fungi, including *B. cinerea* [30,31]. Therefore, isolate 67-1 was selected for the following pot experiment. According to our data, the control effect of *C. rosea* alone was only slightly lower than the application of fungicides, which further proved that *C. rosea* 67-1 isolate is a promising BCA against *B. cinerea*. As *C. rosea* acts by competing for space and nutrients in wounded tissues [46], its efficacy in colonizing the host may depend on the amount of conidia applied. According to Borges et al., who compared the conidial concentration and disease control, the best results for control were obtained at a concentration above 10^6^ conidia mL^−1^ one day before or simultaneously with the pathogens on tomato plants [15]. Thus, we applied *C. rosea* at 10^7^ conidia mL^−1^ concentration for the control of *B. cinerea* in our pot experiments and halved the concentration of *C. rosea* to 5 × 10^6^ conidia mL^−1^ when combined with the fungicides. Based on Chatterton and Punja’s research, environmental factors such as temperature and pH were major factors that influenced population levels of *C. rosea* [14,47]. The optimum temperature for leaf colonization was 20–25 °C, and maximum population densities on the leaves required at least 12 h of continuous leaf wetness [14]. Hence, greenhouse environmental conditions were maintained at 90% relative humidity and 25 °C room temperature for the pot experiment to obtain a stable and efficient control effect.

Our study showed a significant synergistic effect of *C. rosea* with SDHIs. The control effect of the combination of *C. rosea* with fluxapyroxad or fluopyram against tomato gray mold was significantly increased compared to that of BCA or SDHI fungicide alone in combination treatment and rotation treatment; the combination allows a two-fold reduction of both the fungicide and BCA dose. Several possible mechanisms for the synergistic effects were observed upon the combined application of *C. rosea* and SDHIs. Firstly, as the primary biological control mechanism, *C. rosea* could secrete cell-wall-degrading enzymes (CWDEs) to degrade the cell wall of the host fungus [48,49,50]. Thus, with the lack of an essential barrier for cell protection, the gray mold might become more vulnerable to the fungicides treated. Second, *C. rosea* produced secondary metabolites such as antibiotics and toxins [51,52], and the combined application of these antibiotics or toxins with SDHIs may show the same synergistic effects as the synergistic effect shown in a combination of fungicides with one another. Third, treating *B. cinerea* infection with *C. rosea* has been reported to induce several defense mechanisms in tomatoes, including fortifying the plant cell wall and stimulating the expression of several signaling molecules [16,19,53]. In this way, the resistance of tomato plants to gray mold is enhanced when inoculated with *C. rosea*. After the fungicide treatment, the plants are less susceptible to gray mold, showing a synergistic effect. 

Whether the BCAs survive on plants or colonize the plants successfully after the application is a crucial step for the biological control activity of many BCAs. Rapid activity loss is thought to be the main reason some BCAs are not successful in the field but show excellent performance in the lab [54]. It is reported that *B. subtilis* was rapidly lost 3 days after application on rice by using real-time qPCR detection [36]. This result is in accordance with a study of *B. subtilis* on a strawberry based on next-generation sequencing [55]. In terms of *C. rosea*, it was confirmed that *C. rosea* could successfully colonize the foliage of geraniums and the roots of cucumbers by using a GUS-transformed isolate, demonstrating the endophytic ability of *C. rosea* in foliar and root tissues [14,47]. In this study, DNA of *C. rosea* was directly extracted from tomato plants, and the fungal dynamics were analyzed by real-time qPCR to quantify *C. rosea* DNA. Although DNA extraction included dead and inactive fungi and may result in a higher gene expression level, it was believed to be the most available method because of its convenience and accuracy [36]. In the qPCR assays, though there were variations between the replicates, the replicates showed a similar trend (Figure 5). Because the absolute quantifications of *B. cinerea* and *C. rosea* were tested, it was very hard to repeat the absolute copy number from the two independent experiments. The environment and the status of the microorganisms can be slightly different from the two replicates, which ultimately influence the colonization. The qPCR test of *C. rosea* demonstrated that *C. rosea* could still be detected on tomatoes when used alone and mixed with fungicides. The qPCR test of *B. cinerea* showed that *C. rosea* and SDHI fungicide significantly reduced the biomass of *B. cinerea*. Compared to the control, the biomass of *B. cinerea* was the lowest in the combination treatment of *C. rosea* and SDHI fungicide, which is consistent with the control efficacy in the greenhouse.

In conclusion, our study showed that *C. rosea* isolates could tolerate high concentrations of SDHIs with no adverse growth effects, suggesting that they were fully compatible with these fungicides. Pot experiment and qPCR assays showed a significant synergistic effect of *C. rosea* with SDHIs in controlling tomato gray mold. These results showed that combining BCA with SDHIs may meet the demands of the Chinese government’s “low fertilizer and low pesticides” campaign. Additional field trials and investigations to monitor the behavior of *C. rosea* in the field can help to determine the optimal timing and the method of this BCA application to control gray mold in tomato production. 

## Figures and Tables

**Figure 1 microorganisms-11-00020-f001:**
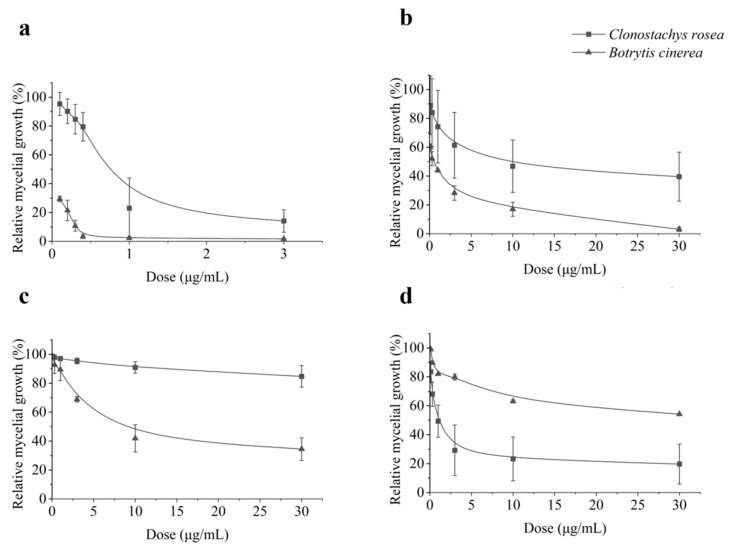
Contrasting in vitro relative growth of *Clonostachys rosea* and *Botrytis cinerea* on different concentrations of (**a**) carbendazim, (**b**) tebuconazole, (**c**) boscalid, and (**d**) pyraclostrobin in PDA medium. Mean and standard deviation from the average of eight *C. rosea* isolates (isolates JLB-7-1, 67-1, SYP-4-2, SHW-1-1, YJS-3-2, GS6-1, NHH-48-2, and BD-2-1) and two *B. cinerea* (YN80 and YN81) isolates were collected.

**Figure 2 microorganisms-11-00020-f002:**
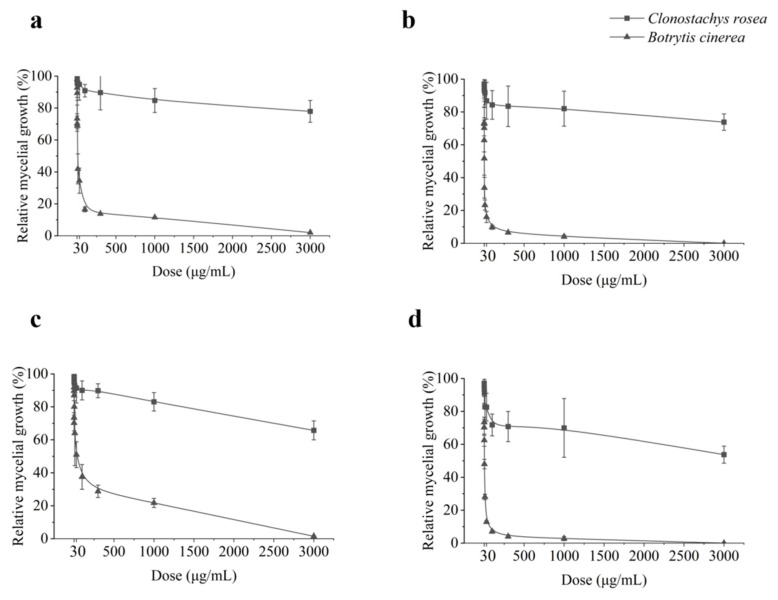
Contrasting in vitro relative growth of *Clonostachys rosea* and *Botrytis cinerea* on different concentrations of (**a**) boscalid, (**b**) fluxapyroxad, (**c**) fluopimomide, and (**d**) fluopyram. Mean and standard deviation from the average of eight *C. rosea* isolates (isolates JLB-7-1, 67-1, SYP-4-2, SHW-1-1, YJS-3-2, GS6-1, NHH-48-2, and BD-2-1) and two *B. cinerea* isolates (YN80 and YN81) were collected.

**Figure 3 microorganisms-11-00020-f003:**
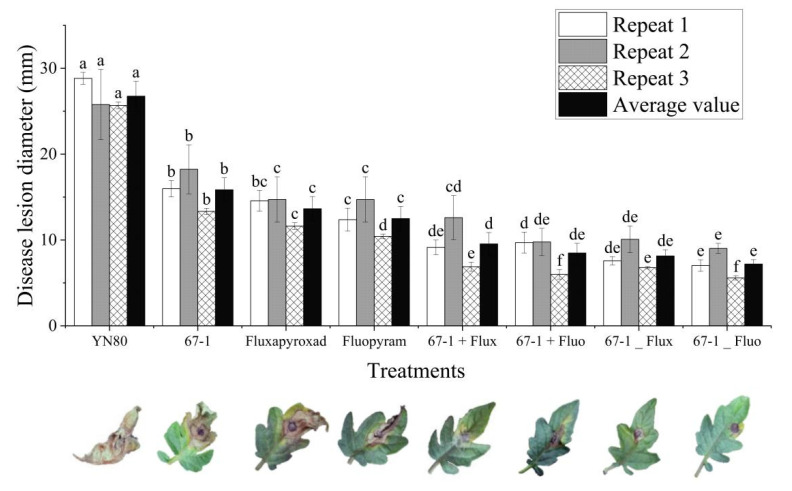
The disease lesion diameter of tomato gray mold when treated by *Clonostachys rosea* 67-1, fluxapyroxad, fluopimomide alone, in combination, or in rotation. Data are presented as the mean ± standard deviation (SD). The different lowercase letters indicate significant differences between different treatments in each repeat at the 5% level of probability. “YN80”, only inoculated with mycelial plugs of *B. cinerea* isolate YN80; “67-1”, sprayed with conidia suspension of *C. rosea* isolate 67-1; “Fluxapyroxad”, sprayed with fluxapyroxad; “Fluopyram”, sprayed with fluopyram; “67-1+Flux”, sprayed with the mixture of 67-1 conidia suspension and fluxapyroxad; “67-1+Fluo” sprayed with the mixture of 67-1 conidia suspension and fluopyram; “67-1_Flux”, sprayed the 67-1 conidia suspension first and fluxapyroxad 24 h later; “67-1_Fluo” sprayed the 67-1 conidia suspension first and fluxapyroxad 24 h later.

**Figure 4 microorganisms-11-00020-f004:**
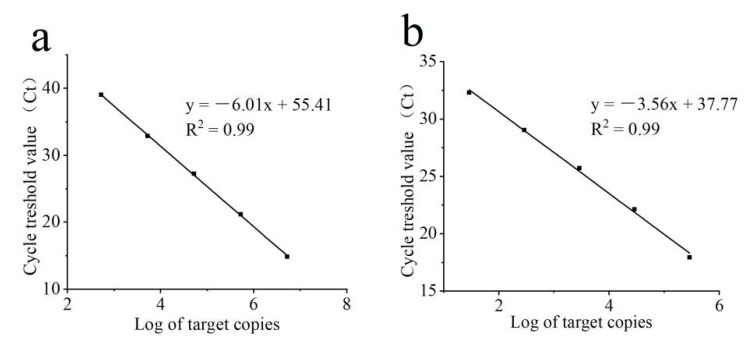
Standard curve of the qPCR for quantification of *Botrytis cinerea* (**a**) and *Clonostachys rosea* (**b**). qPCR standard regression was obtained from the log of the copy number of *B. cinerea* (**a**) and *C. rosea* (**b**) against the corresponding cycle threshold (Ct) values. Target range was from 5.25 × 10^2^ to 5.25 × 10^6^ copies per reaction for *B. cinerea*, and 2.91 × 10 to 2.91 × 10^5^ per reaction for *C. rosea*. The number of target copies on a log-scaled X-axis were plotted against Ct values from 14 to 40 for *B. cinerea* isolate YN80 and 14 to 32 for *C. rosea* isolate 67-1 on the Y-axis. Linear regression equation of the *B. cinerea* standard curve was Y = –6.01x + 55.41 at R^2^ = 0.99. Linear regression equation of the *C. rosea* standard curve was Y = –3.56x + 37.77 at R^2^ = 0.99.

**Figure 5 microorganisms-11-00020-f005:**
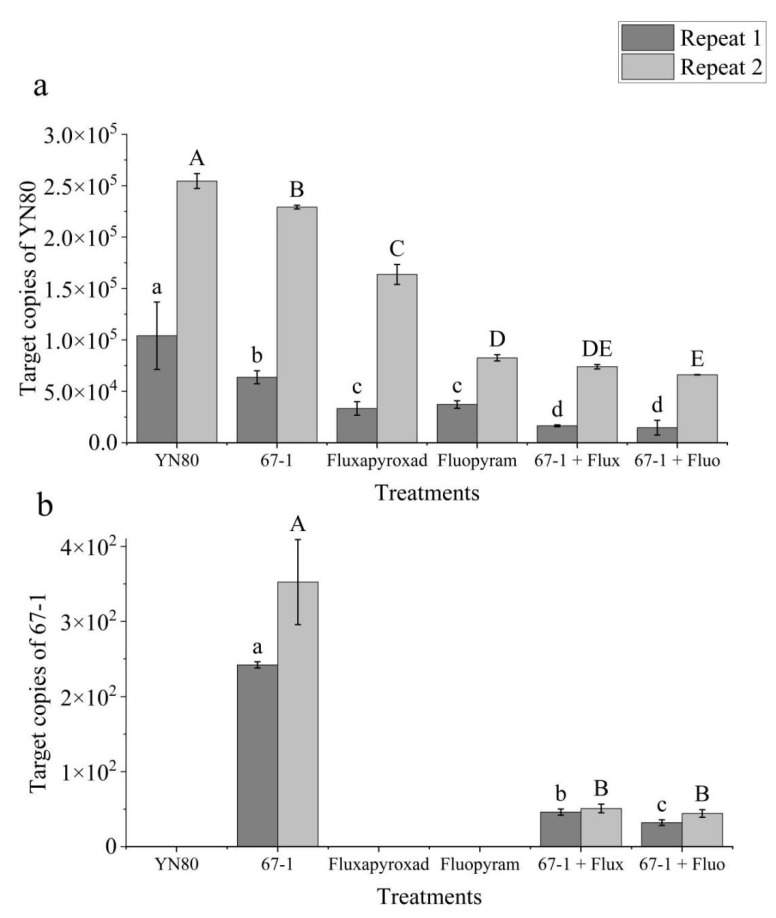
Quantification of *Botrytis cinerea* (**a**) and *Clonostachys rosea* (**b**) by TaqMan qPCR in treated tomato plants after 10 days under greenhouse conditions. Data are presented as the mean ± standard deviation (SD). Uppercase letters and lowercase letters represent two independent repeated tests. The different letters indicate significant differences between different treatments in each repeat (α = 0.05). “YN80”, only inoculated with mycelial plugs of *B. cinerea* isolate YN80; “67-1”, sprayed with conidia suspension of *C. rosea* isolate 67-1; “Fluxapyroxad”, sprayed with fluxapyroxad; “Fluopyram”, sprayed with fluopyram; “67-1+Flux”, sprayed with the mixture of 67-1 conidia suspension and fluxapyroxad; “67-1+Fluo” sprayed with the mixture of 67-1 conidia suspension and fluopyram.

**Table 1 microorganisms-11-00020-t001:** Fungicide sensitivities of the *Clonostachys rosea* and *Botrytis cinerea* isolates to carbendazim, tebuconazole, pyraclostrobin, boscalid, fluxapyroxad, fluopimomide, and fluopyram.

Species	Isolate	EC_50_ (μg mL^−1^) ^z^ ± SE						
		Carbendazim	Tebuconazole	Pyraclostrobin	Boscalid	Fluxapyroxad	Fluopimomide	Fluopyram
*Clonostachys rosea*	JLB-7-1	1.66 ± 0.27	0.02 ± 0.01	11.17 ± 2.08	>1000	>1000	>1000	>1000
	67-1	1.04 ± 0.66	10.24 ± 1.71	0.52 ± 0.29	>1000	>1000	>1000	>1000
	SYP-4-2	0.34 ± 0.23	9.39 ± 1.63	0.59 ± 0.32	>1000	>1000	>1000	>1000
	SHW-1-1	0.73 ± 0.48	21.11 ± 11.20	4.46 ± 3.42	>1000	>1000	>1000	>1000
	YJS-3-2	0.50 ± 0.34	20.39 ± 14.54	0.74 ± 0.35	>1000	>1000	>1000	>1000
	GS6-1	0.89 ± 0.57	7.46 ± 5.95	0.52 ± 0.31	>1000	>1000	>1000	>1000
	NHH-48-2	0.71 ± 0.49	102.86 ± 53.70	3.08 ± 1.31	>1000	>1000	>1000	>1000
	BD-2-1	0.74 ± 0.52	16.87 ± 11.57	0.66 ± 0.47	>1000	>1000	>1000	>1000
*Botrytis cinerea*	YN80	0.01 ± 0.002	0.27 ± 0.16	31.95 ± 10.97	15.46 ± 4.50	1.75 ± 1.41	12.96 ± 5.85	1.12 ± 0.67
	YN81	0.03 ± 0.006	0.47 ± 0.17	22.69 ± 6.16	5.95 ± 3.98	0.40 ± 0.33	33.41 ± 7.34	1.92 ± 1.15

^z^ EC_50_ = Effective concentration that inhibits 50% of fungal growth. SE, standard error.

**Table 2 microorganisms-11-00020-t002:** In vitro germination rate of conidia of *Clonostachys rosea* and *Botrytis cinerea* under different fungicide concentrations.

Species ^y^	Gemination Rate of Conidium at Different Fungicide Concentrations (%) ^z^						
	Fungicide	Concentrations of Fungicides (µg mL^−1^)					
		0	7.5	15	30	60	120
*Clonostachys rosea*	Fluxapyroxad	99.99 ± 0.01	99.99 ± 0.01	99.99 ± 0.01	99.99 ± 0.01	99.99 ± 0.01	99.99 ± 0.01
	Fluopyram	99.99 ± 0.01	99.99 ± 0.01	99.99 ± 0.01	99.34 ± 0.47	97.70 ± 0.07	97.34 ± 0.41
*Botrytis cinerea*	Fluxapyroxad	95.71 ± 0.71	16.95 ± 1.86	8.34 ± 3.36	5.67 ± 3.26	3.61 ± 1..02	1.80 ± 0.08
	Fluopyram	95.71 ± 0.71	21.05 ± 0.76	8.62 ± 3.71	10.81 ± 0.50	4.62 ± 0.37	2.00 ± 0.08

^y^ Isolate 67-1 represented the *Clonostachys rosea,* isolate YN80 represented the *Botrytis cinerea.* ^z^ Mean ± standard deviation; at least 200 conidia were examined microscopically to determine germination in each of three replicate plates 24 h at 22 °C.

**Table 3 microorganisms-11-00020-t003:** Control efficacy on tomato gray mold in greenhouse experiment.

Treatments ^y^	Control Efficacy ^z^
67-1	46.42% ± 3.14% d
Fluxapyroxad	52.28% ± 4.17% c
Fluopyram	58.31% ± 3.57% c
67-1+Flux	70.91% ± 3.65% b
67-1+Fluo	71.94% ± 6.34% ab
67-1_Flux	73.65% ± 1.24% ab
67-1_Fluo	77.07% ± 2.26% a

^y^ Treatment “67-1”, sprayed with conidia suspension of *Clonostachys rosea* isolate 67-1; “Fluxapyroxad”, sprayed with fluxapyroxad; “Fluopyram”, sprayed with fluopyram; “67-1+Flux”, sprayed with the mixture of 67-1 conidia suspension and fluxapyroxad; “67-1+Fluo” sprayed with the mixture of 67-1 conidia suspension and fluopyram; “67-1_Flux”, sprayed the conidia suspension of *C. rosea* isolate 67-1 first and fluxapyroxad 24 h later; “67-1_Fluo” sprayed the conidia suspension of *C. rosea* isolate 67-1 first and fluxapyroxad 24 h later. After 24 h, all of the above tomato seedlings treatments were inoculated with 5-mm-agar plugs of *Botrytis cinerea* isolate YN80 on the leaves. ^z^ Data are presented as the mean ± standard deviation (SD). The different lowercase letters indicate significant differences between different treatments in each repeat at the 5% level of probability. One-way analysis of variance (ANOVA) with a least significant difference (LSD) test in SPSS software (version 21.0; SPSS Inc.) was used to evaluate the significant differences between treatments.

## Data Availability

The data used to support the findings of this study are available from the corresponding authors upon request.

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
