# Peer review of "Synergistic Effects of Clonostachys rosea Isolates and Succinate Dehydrogenase Inhibitors Fungicides against Gray Mold on Tomato"

_microorganisms, 2022, doi:10.3390/microorganisms11010020_

Round 1

Reviewer 1 Report

Synergistic effects of Clonostachys rosea isolates and succinate dehydrogenase inhibitors fungicides against gray mold on tomato

This manuscript describes a strategy based on the combination between a BCA and a SDHI to control B. cinerea in tomato plants. Combination treatments halved doses of each component relative to treatments where they are applied alone. In the Materials and Methods section is convenient to describe the statistical analysis used.

 The work is well achieved although there are some points that need to be clarified or explained as listed below:

 Lines 27-28. Define BCA when first mentioned.

Line 39, 85, 307 and others: Verify that all scientific names are properly written using italics fond

Line 44-45 and others. High genetic variability found in B. cinerea could lead to different susceptibility of strains to the proposed strategy. Did you test the combination with different B. cinerea strains?

Line 104: 10 to the 4th? Please verify.

Line 114: Petri with capital "P"

Lines 125-128: Why did each fungus grow on different media? Please explain.

Lines 134-136: Only two repetitions? This is a drawback in the experimental design.

Line 203: CT or Ct?

Lines 217-218: Define MBC, DMI and QoI as done previously with SHDI

Discussion and lines 235-240: Why are C. rosea and B. cinerea so different in terms of tolerance to the same fungicides? Any mechanisms already described?

Lines 288-289: In fact, combination of two BCA is also a probed strategy, see Biocontrol Science and Technology, 31:10, 1080-1097, DOI: 10.1080/09583157.2021.1923656

Line 540 and others along the document: Write "in vitro" with italics fond

Discussion.

Are there evidences indicating that the combination of both chemical SDHI and C. rosea act on fungal strains showing already resistance to SDHI?

Author Response

Review 1:

This manuscript describes a strategy based on the combination between a BCA and a SDHI to control B. cinerea in tomato plants. Combination treatments halved doses of each component relative to treatments where they are applied alone. In the Materials and Methods section is convenient to describe the statistical analysis used.

 The work is well achieved although there are some points that need to be clarified or explained as listed below:

Response: Thank you very much for your comments and suggestions to improve the manuscript.

Lines 27-28. Define BCA when first mentioned.

Line 39, 85, 307 and others: Verify that all scientific names are properly written using italics fond

Response: Revised.

Line 44-45 and others. High genetic variability found in B. cinerea could lead to different susceptibility of strains to the proposed strategy. Did you test the combination with different B. cinerea strains?

Response: In the experiment of fungicides sensitivity assessments of Botrytis cinerea in vitro, two isolates YN80 and YN81 were used. They were sensitive to all SDHI fungicides. The effectiveness of SDHI fungicides on B. cinerea has been reported in several studies. Also, the SDHIs have already been registered for the control of gray mold in several countries (https://www.frac.info/docs/default-source/publications/frac-code-list/frac-code-list-2022--final.pdf?sfvrsn=b6024e9a_2.). Besides, according to previous studies, Clonostachys rosea has been shown to be effective in controlling gray mold in several crops, both in field and greenhouse cultivations [12-15].

Line 104: 10 to the 4th? Please verify.

Response: Should be 10 to the 5th, thanks for correcting.

Line 114: Petri with capital "P"

Response: Revised.

Lines 125-128: Why did each fungus grow on different media? Please explain.

Response: Czapek Dox Liquid Medium was used for sporulation of Clonostachys rosea, while C. rosea produce few conidia on PDA. 

Lines 134-136: Only two repetitions? This is a drawback in the experimental design.

Response: The results of the two repetitions were consistent, so we did not do more repetitions.

Line 203: CT or Ct?

Response: We revised it as “Ct”.

Lines 217-218: Define MBC, DMI and QoI as done previously with SHDI

Response: Revised.

Discussion and lines 235-240: Why is C. rosea and B. cinerea so different in terms of tolerance to the same fungicides? Any mechanisms already described?

Response: So far, the inherent resistance mechanisms in C. rosea remained unknown. The natural resistance of fungus to SDHI fungicides is not uncommon. Colletotrichum spp. and Fusarium fujikuroi have been reported to be naturally resistant to SDHI fungicides. But the mechanisms have not been studied before. We have discussed the corresponding contents in the 2nd paragraph of the Discussion.

Lines 288-289: In fact, a combination of two BCA is also a probed strategy, see Biocontrol Science and Technology, 31:10, 1080-1097, DOI: 10.1080/09583157.2021.1923656

Response: We added the reference and relevant content.

Line 540 and others along the document: Write "in vitro" with italics fond

Response: Revised.

Discussion.

Are there evidences indicating that the combination of both chemical SDHI and C. rosea act on fungal strains showing already resistance to SDHI?

Response: There were no relevant literature to our knowledge, and we have not conducted this experiment. You gave us a very good suggestion. We suppose that the combination of SDHI and C. rosea can act on SDHI-resistant plant pathogenic fungi because C. rosea was highly efficient in controlling many plant pathogenic fungi when used alone. Next, we will further confirm this assumption.

Reviewer 2 Report

In this manuscript, Song et al. present their research on the synergistic effects of C. rosea isolates and succinate dehydrogenase inhibitors (SDHI) against the pathogen B. cinerea. The study deals with an interesting aspect with both scientific and economic components. The language is, generally clear and the whole issue is adequately introduced. However, due to the obscurity of data analysis, I cannot explicitly conclude on the magnitude of the significance of the results.

Firstly, the authors do not provide the methodology for the EC50 estimation. Furthermore, the significant tests used for Figures 2 and 4 are not presented in Methods, and experimental replications are not clear if/how they are modeled. A linear (mixed) model approach might prove very conclusive, in this case. Moreover, those experimental replicates show high levels of variation (especially in the case of qPCR assays), a fact that the authors do not discuss at all. These big experimental fluctuations combined with the unclear data analysis methodology render the significance of the treatment tests problematic.

I would strongly suggest the authors to address the aforementioned points before considering the possibility of publishing this work in Microorganisms.

Author Response

Review 2:

In this manuscript, Song et al. present their research on the synergistic effects of C. rosea isolates and succinate dehydrogenase inhibitors (SDHI) against the pathogen B. cinerea. The study deals with an interesting aspect with both scientific and economic components. The language is, generally clear and the whole issue is adequately introduced. However, due to the obscurity of data analysis, I cannot explicitly conclude on the magnitude of the significance of the results.

Firstly, the authors do not provide the methodology for the EC50 estimation.

Respond: We feel great thanks for your professional review work on our article. EC50 values were calculated as below: “Fungicide sensitivity, as measured by the 50% effective concentration (EC50) value, was calculated as described by Wong and Wilcox (2002). Briefly, the percent relative growth (RG) was calculated as (radial growth at fungicide concentration/radial growth on the non-amended check plate) × 100. The EC50 value was estimated by linear regression of the probit-transformed relative inhibition (RI) value (RI = 1 − RG) on log10 transformed-fungicide concentration. The EC50 value for each isolate was calculated as the mean of the three replicates.” The corresponding contents were added in section 2.2.

.  

Furthermore, the significant tests used for Figures 2 and 4 are not presented in Methods, and experimental replications are not clear if/how they are modeled. A linear (mixed) model approach might prove very conclusive, in this case.

Respond: Fig. 2 was mentioned in section 3.1, and the corresponding methods were mentioned in section 2.2, “the percent relative growth (RG) was calculated as (radial growth at fungicide concentration/radial growth on the non-amended check plate) × 100.”. The experiments were repeated three times as mentioned in section 2.2: “The EC50 value for each isolate was calculated as the mean of the three replicates.”

The methods for Fig. 4 were presented in section 2.5: “Standard curves were generated by plotting the logarithmic values of target copies versus the corresponding cycle threshold (Ct) values”. Each standard curve analysis was measured in three technical replicates and the independent tests gave highly similar results. A linear fitting was used for the model here. The corresponding contents were added in line 199-202. 

Moreover, those experimental replicates show high levels of variation (especially in the case of qPCR assays), a fact that the authors do not discuss at all.

Respond: Yes, the replicates showed variations in the qPCR assays, that’s why we did not combine the data from the replicates. But as you can see, the replicates showed roughly a similar trend in both Fig. 5a and Fig. 5b. Since we were testing the absolute quantification of Botrytis cinerea and Clonostachys rosea, it was very hard to repeat the absolute copy number from the two independent experiments. The environment, and the status of the microorganisms can be slightly different from the two replicates, which can ultimately influence colonization. We discussed the contents in the 5th paragraph of the Discussion according to your suggestion. 

These big experimental fluctuations combined with the unclear data analysis methodology render the significance of the treatment tests problematic.

I would strongly suggest the authors address the aforementioned points before considering the possibility of publishing this work in Microorganisms.

Respond: we hope the revised manuscript could be acceptable to you. 

Reviewer 3 Report

Dear Editors and Authors

The publication contains very useful information. I have included a few corrections concerning mainly spelling in the attached file. I believe that the work is valuable and should be published with minor corrections.

Author Response

Review 3:

Comments and Suggestions for Authors

Dear Editors and Authors

The publication contains very useful information. I have included a few corrections concerning mainly spelling in the attached file. I believe that the work is valuable and should be published with minor corrections.

  1. Line 39, Botrytis cinerea should bein italics.
  2. Line 40, “vegetable” should be “vegetables”.
  3. Line 58, “growth” should be “development”.
  4. Line 61, “bio fungicide” should be “biofungicide”.
  5. Line 62, “disease” should be “diseases”, “have” should be “has”.
  6. Line 72, give literatures.
  7. Line 77, “were” should be “was”.
  8. Line 85, “flupyram” should be “fluopyram”, “C. rosea” should bein italics.
  9. Line 89, “Clonostachys rosea” should be “ rosea”.
  10. Line 90, “isolate” should be “isolates”.
  11. Line 98, delete the space.
  12. Line 108, 116, 122 and 125, “Clonostachys rosea and Botrytis cinerea” requires abbreviations.
  13. Line 126, The composition of Czapek Dox medium can be delated. It is basic medium used in microbiology.
  14. Line 138, Give tomato cultivar.
  15. Line 142, The treatments should be presented more clearly but also shorter. You should not repeat e.g"isolate" "67-1" - you used only 67-1 isolate. Maybe any abbrevitations would be a good way.
  16. Line 166, The composition is obvious. Delete it.
  17. Line 214, abbreviation.
  18. Line 216, “compatibilty” should be “compatibility”.
  19. Line 225, “were” should be “was”.
  20. Line 226, Are the data presented anywhere? If not, add "Data not presented"
  21. Line 238, “flupyram” should be “fluopyram”.
  22. Line 250 and 252, delete "inoculated", Inoculated by what?
  23. Line 263, correct 107.
  24. Line 264, abbreviation.
  25. Line 301, without hyphen.
  26. Line 307, italic.
  27. Line 521, 523, 528, 540, 545, 551, 555, 566, 571, 572, 580, abbreviation.

Response: Thank you very much for your comments and suggestions to improve the manuscript. We revised these questions based on your suggestions.

Reviewer 4 Report

The topic is interesting, and appropriate for the Special Issue (Advanced Research on Biological Control of Plant Disease or Microbial Interactions). The results are interesting, and provide useful information about the combined application of BCA C. rosea strain with determined fungicide against B. cinerea. Beyond in vitro experiments, in vivo experiments also support the possibility of the effective combined application.

However, the manuscript must be improved.

1)      Materials and methods part

·         References about the tested microbes should be provided (e.g. there are references [40, 41] about C. rosea isolate 67-1 in the Discussion part: 40, 41.

·         Media and solutions compositions should be in g L-1 format.

·         The type of water used for preparing media or solutions should be given (e.g. tap water, deionized water, distilled water, nuclease-free water).

·         PDA preparation should be given, if it was prepared from potato (and not fom ready to prepare dehydrated powder medium).

·         The producers of the chemicals should be provided (e.g. Rows 132-133 bact-peptone, sodium acetate yeast extract, etc.).

·         Please correct the producer of fluxapyroxad (row 101).

·         Provide reference for the preparation of fungicide amended media, or describe the preparation in more detailed.

·         Why SHAM was added to added to pyraclostrobin-amended PDA? Provide information and reference, as well as producer of SHAM.

·         Reference should be added for EC50 calculation.

·         Calculation of Relative mycelial growth (%) indicated in Figures 1 and 2 should be described.

·         More details should be provided about the tested tomato seedlings – please give the full current name and the cultivar.

·         Provide age/development status of the tomato seedlings when inoculation and/or fungicide application was performed.

·         Please give a reference or more details about the inoculation of tomato seedlings with mycelial plugs.

·         Calculation details or reference for disease incidence (Row 162) should be provided. Results of this parameter is not provided in the further chapter.

·         Determination and calculation details of Control efficacy in Table 3 is missing.

·         Chapter 2.5 is difficult to follow. Please give first the details for the preparation and measurement of DNA standard. Than the details for the sample preparation (tomato leaves), finally the details of qPCR detection of C. rosea and B. cinerea DNA. Further problems: two PCR amplification profiles were given for the same primer pairs (Rows 181-182 and 186-187)

·         The mode of detection for qPCR analysis is missing (name and producer of dye, detection wavelength).

·         Statistical analysis chapter should be added.

2)      Results part

·         Chapters 3.1 and 3.2. are about mycelial growth inhibition. The two chapters should be merged, and mycelial growth inhibition should be indicated in the titles. (e.g. In vitro mycelial growth inhibition of C. rosea and B. cinerea by different fungicides

·      The qPCR assay examined only the DNA, therefore it could not provide information about the survival of C. rosea among different conditions. Higher target copies (Figure 5b) indicates better growth comparing to lower target copies. However, there is no information, if those DNA are from living or dead fungi. Please modify the description of your results and discussion.

3)      Abbreviations

·         Provide the full name/detailed description of all the abbreviation at their first mention (e.g. a.i.; MBC, DMI, QoI)

4)      Manuscript format

·         Unify the size of the letters (e.g. 290-295)

·         Genus and species names should be in Italics (e.g. Botrytis cinerea – Row 39; Trichoderma – Row 307). Full species name should be in short form following the first mention.

·         Superscript is missed (e.g. 107 ml L-1 and not 107 spores/mL – Row – 263)

5)      Tables and Figures

·         Tables 1 and 2: use Species and not Taxonomic

·         Table 1 and 2: Isolate and Fungicide (not plural form); Mind the name of the fungicides in one row

·         Table 3: provide details for statistical analysis.

·         Table 3, Figures 2 and 4: Give fungal species name and indicate strain number in footnotes.

Author Response

Review 4:

Comments and Suggestions for Authors

The topic is interesting, and appropriate for the Special Issue (Advanced Research on Biological Control of Plant Disease or Microbial Interactions). The results are interesting, and provide useful information about the combined application of BCA C. rosea strain with determined fungicide against B. cinerea. Beyond in vitro experiments, in vivo experiments also support the possibility of the effective combined application.

However, the manuscript must be improved.

Response: Thank you very much for your kind comments and suggestions.

1)      Materials and methods part

  • References about the tested microbes should be provided (e.g. there are references [40, 41] about C. rosea isolate 67-1 in the Discussion part: 40, 41.

Response: We added the references based on your suggestion.

  • Media and solutions compositions should be in g L-1format.

Response: Revised.

  • The type of water used for preparing media or solutions should be given (e.g. tap water, deionized water, distilled water, nuclease-free water).

Response: We revised it as “deionized water”.

  • PDA preparation should be given, if it was prepared from potato (and not fom ready-to-prepare dehydrated powder medium).

Response: We revised it as “All isolates used for inoculations were maintained on PDA (Potato Dextrose Agar: potato 200 g L-1, glucose 15 g L-1, agar 15 g L-1, add deionized water to 1 L. The fresh potato was boiled in deionized water for 20 minutes, then filtered and the potato juice was taken to make medium) medium at 25℃ in darkness unless otherwise specified.”

  • The producers of the chemicals should be provided (e.g. Rows 132-133 bact-peptone, sodium acetate yeast extract, etc.).

Response: We added the producer.

  • Please correct the producer of fluxapyroxad (row 101).

Response: it should be BASF Corp., Research Triangle Park, NC.

  • Provide reference for the preparation of fungicide amended media, or describe the preparation in more detail.

Response: We added the reference.

  • Why SHAM was added to the pyraclostrobin-amended PDA? Provide information and reference, as well as the producer of SHAM.

Response: We added the reference and producer.

  • Reference should be added for EC50 calculation.

Response: We added the reference.

  • The calculation of Relative mycelial growth (%) indicated in Figures 1 and 2 should be described.

Response: We added it in section 2.2 as “the percent relative growth (RG) was calculated as (radial growth at fungicide concentration/radial growth on the non-amended check plate) × 100.”

  • More details should be provided about the tested tomato seedlings – please give the full current name and the cultivar.

Response: The cultivar is Jinpengwuxian. They were purchased from Xi’an Jinpeng Seedling Co., Ltd. and transplanted to the pots. The information was added in section 2.4. 

  • Provide age/development status of the tomato seedlings when inoculation and/or fungicide application was performed.

Response: Forty-day-old tomato seedlings were used. The information was added in section 2.4.

  • Please give a reference or more details about the inoculation of tomato seedlings with mycelial plugs.

Response: We added the reference.

  • Calculation details or references for disease incidence (Row 162) should be provided. Results of this parameter is not provided in the further chapter.

Response: We did not calculate the disease incidence. Sorry for this mistake, and we revised it as “the control efficacy”.

  • The determination and calculation details of Control efficacy in Table 3 is missing.

Response: We revised it in section 2.4 and 2.6 as “Seven days after inoculation, lesion diameters were measured at two perpendicular directions using a caliper, and the control efficacy of each treatment was calculated.” (line 179-181) and “Control efficacy = [(lesion diameter of the control – lesion diameter of the treatment)/lesion diameter of the control] × 100% (line 232-234).

  • Chapter 2.5 is difficult to follow. Please give first the details for the preparation and measurement of DNA standard. Then the details for the sample preparation (tomato leaves), and finally the details of qPCR detection of C. rosea and B. cinerea DNA. Further problems: two PCR amplification profiles were given for the same primer pairs (Rows 181-182 and 186-187)

Response: Thanks! We revised chapter 2.5 according to your suggestion. The PCR amplification profiles were corrected. 

  • The mode of detection for qPCR analysis is missing (name and producer of dye, detection wavelength).

Response: 50×ROX Reference Dye II purchased from Tsingke Biotechnology Co., Ltd. Was used in the study. The detection wavelength was 520 nm ±10 nm. The information was added in Line 227.

  • A statistical analysis chapter should be added.

Response: We added the Statistical analysis chapter as “2.6. Statistical analysis. Relative mycelial growth (%) = (diameter of the treatment - diameter of the plug) / (diameter of the control- diameter of the plug) × 100. Control efficacy = [(lesion diameter of the control – lesion diameter of the treatment)/lesion diameter of the control] × 100%. Results were represented as the mean values ± standard deviation. One-way analysis of variance (ANOVA) with a least significant difference (LSD) test in SPSS software (version 21.0; SPSS Inc.) was used to evaluate the significant differences between treatments.”

2)      Results part

  • Chapters 3.1 and 3.2. are about mycelial growth inhibition. The two chapters should be merged, and mycelial growth inhibition should be indicated in the titles. (e.g. In vitro mycelial growth inhibition of C. roseaand B. cinerea by different fungicides)

Response: We combined chapters 3.1 and 3.2 according to your suggestion, and changed the title to in vitro mycelial growth inhibition of C. rosea and B. cinerea by different fungicides.

  • The qPCR assay examined only the DNA, therefore it could not provide information about the survival of C. rosea among different conditions. Higher target copies (Figure 5b) indicates better growth comparing to lower target copies. However, there is no information, if those DNA are from living or dead fungi. Please modify the description of your results and discussion.

Response: Yes, thanks for correcting that, the DNA tested by qPCR tests included dead and inactive fungi. According to your suggestion, we changed our statement to “the qPCR results showed that C. rosea could still be detected on tomato plants when mixed with fluopyram and fluxapyroxad” in both results part and discussion part. We changed “survival” to “biomass”.

3)      Abbreviations

  • Provide the full name/detailed description of all the abbreviations at their first mention (e.g. a.i.; MBC, DMI, QoI)

Response: Revised.

4)      Manuscript format

  • Unify the size of the letters (e.g. 290-295)

Response: We unified the size of the letters. It was a problem with system typesetting.

  • Genus and species names should be in Italics (e.g. Botrytis cinerea – Row 39; Trichoderma– Row 307). Full species name should be in short form following the first mention.

Response: Revised.

  • Superscript is missed (e.g. 107ml L-1 and not 107 spores/mL – Row – 263)

Response: Revised.

5)      Tables and Figures

  • Tables 1 and 2: use Species and not Taxonomic
  • Table 1 and 2: Isolate and Fungicide (not plural form); Mind the name of the fungicides in one row
  • Table 3: provide details for statistical analysis.

Response: Revised.

  • Table 3, Figures 2 and 4: Give fungal species names and indicate strain number in footnotes.

Response: Modified.

Round 2

Reviewer 1 Report

The manuscript has been sufficiently improved to warrant publication in Microorganisms.

Author Response

Thanks

Reviewer 2 Report

The authors' responses are satisfying.

Author Response

Thanks.

Reviewer 4 Report

The manuscript has been improved. The authors corrected the manuscript following my suggestions.

I have got only few minor suggestions:

1) In the following sentence: „Briefly, the percent relative growth (RG) was calculated as (radial growth at fungicide concentration/radial growth on the non-amended check plate)..”

„check plate” suggested to change to „control plate”

·        2) Add the current scientific name in the following sentence: „Tomato (Solanum lycopersicum Dunal L.) seedlings

·       3) Please change all the concentrations in the form recommended by the journal, e.g. μg/mL change to μg mL-1, conidiospores /mL change to conidiospores mL-1

Author Response

1) In the following sentence: „Briefly, the percent relative growth (RG) was calculated as (radial growth at fungicide concentration/radial growth on the non-amended check plate)..”

„check plate” suggested to change to „control plate”

Respond: Modified.

  •      2) Add the current scientific name in the following sentence: „Tomato (Solanum lycopersicum Dunal L.) seedlings

Respond:Modified. 

  •      3) Please change all the concentrations in the form recommended by the journal, e.g. μg/mL change to μg mL-1, conidiospores /mL change to conidiospores mL-1
  • Respond:Modified.